# Gene gain and loss push prokaryotes beyond the homologous recombination barrier and accelerate genome sequence divergence

Jaime Iranzo [1,2], Yuri I. Wolf [1], Eugene V. Koonin [1]* & Itamar Sela [1]*

Bacterial and archaeal evolution involve extensive gene gain and loss. Thus, phylogenetic trees of prokaryotes can be constructed both by traditional sequence-based methods (gene trees) and by comparison of gene compositions (genome trees). Comparing the branch lengths in gene and genome trees with identical topologies for 34 clusters of closely related bacterial and archaeal genomes, we show here that terminal branches of gene trees are systematically compressed compared to those of genome trees. Thus, sequence evolution is delayed compared to genome evolution by gene gain and loss. The extent of this delay differs widely among bacteria and archaea. Mathematical modeling shows that the divergence delay can result from sequence homogenization by homologous recombination. The model explains how homologous recombination maintains the cohesiveness of the core genome of a species while allowing extensive gene gain and loss within the accessory genome. Once evolving genomes become isolated by barriers impeding homologous recombination, gene and genome evolution processes settle into parallel trajectories, and genomes diverge, resulting in speciation.

[1] National Center for Biotechnology Information, National Library of Medicine, National Institutes of Health, Bethesda, MD 20894, USA. [2] Present address: Centro de Biotecnología y Genómica de Plantas, Universidad Politécnica de Madrid (UPM)-Instituto Nacional de Investigación y Tecnología Agraria y Alimentaria (INIA), Campus de Montegancedo-UPM, Pozuelo de Alarcón, 28223 Madrid, Spain. *email: koonin@ncbi.nlm.nih.gov; itamar.sela@nih.gov

Evolution of bacterial and archaeal genomes is a highly dynamic process that involves extensive gain and loss of genes, with turnover rates comparable to if not exceeding the rate of nucleotide substitution[1–3]. Gene gain and loss occur through insertion and deletion of genome segments of variable size, including large genomic islands, via mechanisms of non-homologous recombination, often involving mobile genetic elements[4,5]. The gene gain and loss events can be used to generate 'gene content trees' that reflect the evolution of microbial pan-genomes and complement traditional phylogenetic trees constructed from sequence alignments of highly conserved marker genes. From the gene content trees, a gene turnover clock can be defined. The gene turnover clock ticks at a rate that does not necessarily correlate with the rate of the traditional, sequence-based molecular clock. Actually, as demonstrated below, substantial non-linearity between the two types of trees is observed, and in this work we analyze its likely causes.

Evolution of prokaryotic populations is strongly affected by homologous recombination, which is regarded as a major contributor to maintaining genetic cohesion by preventing sequence divergence via gene conversion[6–9]. Efficient homologous recombination between two genomes requires the presence of (nearly) identical nucleotide sequences flanking the exchanged genomic regions. The minimum length of these flanks depends on the species, with typical values around 25–100 nucleotides[10,11]. As genomes diverge, the probability to find fully conserved flanking sequences decreases, and so does the efficiency of recombination[12,13]. The existence of genetic barriers to homologous recombination was initially observed in experimental studies which have shown that sequence divergence of over 5% can prevent most recombination events in some bacteria[14–16]. Subsequently, comparative genomic analyses have confirmed that barriers to homologous recombination are widespread and can be used to define biological species in bacteria and archaea[17–19]. Mathematical modeling of the molecular processes involved in homologous recombination has shown that barriers to recombination can build up spontaneously and lead to speciation if (i) recombination efficiency decays sufficiently fast with sequence divergence and (ii) the balance of mutation and recombination favors the accumulation of sequence variability in the population[20–24]. Barriers to recombination can also arise after the acquisition of new genes[25–27], especially if the newly acquired genes are involved in niche specialization[28–30]. In this case, the barriers seem to result, in part, from selection against gene conversion events that would lead to the loss of the recently acquired, beneficial genes[27]. Because barriers to recombination do not necessarily affect the whole genome, prokaryotic genomes can diverge at some loci while remaining cohesive at others[25]. Thus, the rates of population divergence and, eventually, speciation depend on the dynamics of recombination barrier emergence and spread across genomes.

Here, we combine comparative genomics and phylogenetic analysis to investigate how the fraction of genes shared by closely related bacterial and archaeal genomes decays with the phylogenetic distance and show that the sequence-based molecular clock and the gene turnover clock are incongruent at short evolutionary times. To elucidate the origin of this discrepancy, we develop a mathematical model of genome evolution that describes the dynamics of sequence divergence in the presence of gene conversion. The model predicts the existence of a recombination-driven delay in the molecular clock, the magnitude of which corresponds to the time required for barriers to recombination to spread across the genome. By fitting the model to genomic data, we obtain estimates of such recombination-driven delays in 34 groups of closely related bacteria and archaea, each covering the whole range of divergence, from early differentiating strains to incipient species. We show that the incongruence between the molecular and gene turnover clocks disappears if the former is corrected to account for the estimated delay. Finally, we investigate the tempo and the factors that contribute to the establishment of barriers to recombination and, eventually, speciation in the populations of diverse bacteria and archaea.

## Results

**Sequence evolution clock lags behind gene turnover clock.** The evolution of gene content in closely related genomes has been recently investigated by mathematical modeling and comparative genomics[31–39]. These analyses have shown that the fraction of genes shared by a pair of genomes decays exponentially with time as the genomes diverge, and that genes can be classified in two categories based on their turnover rates[32]. Here, we extend these approaches to sets of 3 and more genomes. For illustration, let us consider a simple scenario in which all genes have the same turnover rate $\lambda$. The fraction of genes shared by a pair of genomes is then

$$I_2 \sim e^{-\lambda D_2} \qquad (1)$$

where $D_2$ is the total evolutionary tree distance, which in the case of 2 genomes is equal to the sum of the distances from each genome to their last common ancestor. We show in the Methods that a similar formula describes the divergence in gene content for groups of 3 or more genomes. Specifically, the fraction of genes shared by $k$ genomes decays as

$$I_k \sim e^{-\lambda D_k} \qquad (2)$$

where $D_k$ is the total evolutionary distance spanned by those $k$ genomes. Given a phylogenetic tree, the total distance $D_k$ is the sum of branch lengths of the subtree that includes the $k$ genomes. The most notable aspect of this result is that the dynamics of gene content divergence is independent of the number of genomes considered ($k$). As a result, plots of the fraction of shared genes ($I_k$) as a function of the total evolutionary distance ($D_k$) for different sample sizes collapse into a single curve (Fig. 1a). This property remains valid under very general models of gene turnover, including the case where the rates of gene gain and loss differ across gene families (see Methods section and Supplementary Note 1), under the condition that the tree distance is proportional to the rate of the gene turnover clock.

To test this theoretical prediction, we analyzed the profiles of gene sharing in 34 groups of closely related genomes from Bacteria and Archaea, each typically including 2 or 3 closely related species (e.g., *Escherichia coli* and *Salmonella enterica*) and multiple strains per species, from almost identical (<0.001 nucleotide substitutions per site in the core genes) to well differentiated (up to a total tree depth of approximately 0.1–0.2 substitutions per site). As a proxy for the evolutionary time, we used the branch lengths of high-resolution sequence-based phylogenetic trees built from concatenated alignments of single-copy core genes. At least at short evolutionary distances, the branch lengths are supposed to accurately reflect evolutionary time because they are determined, primarily, by synonymous substitutions that are supposed to be effectively neutral and accumulate in a clock-like manner[40]. Then, we sampled subsets of genomes and represented the fraction of shared genes as a function of the total tree distance. At odds with the theoretical expectation, we found that gene-sharing decay curves depend on the number of sampled genomes: as more genomes are added, the curves shift down and the fraction of shared genes becomes smaller than expected (Fig. 1c shows a representative case; see also Supplementary Fig. 1). The shift is equally observed for small and large subsets of sampled genomes ($k$), indicating that this

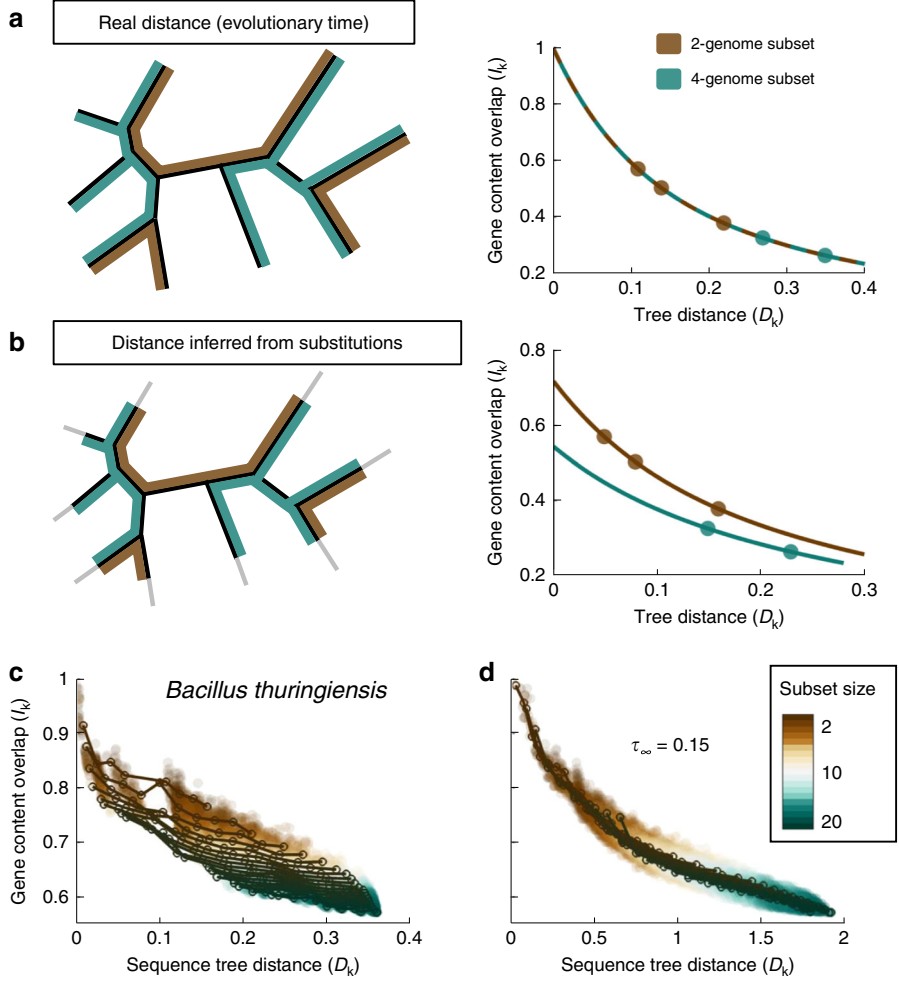

**Fig. 1** Compression of terminal branches of sequence-based phylogenetic trees detected through analysis of gene content decay curves. **a** If tree distances are proportional to the true evolutionary time, the fraction of genes shared by a subset of genomes will decay with the total length of the subtree, and the decay curves will be the same regardless of the number of genomes in the subset. For illustration purposes, three subsets of 2 genomes are highlighted in brown, and two subsets of 4 genomes are highlighted in green. **b** Homologous recombination between pairs of closely related genomes erases recent sequence divergence which results in an underestimation of the evolutionary times associated with terminal tree branches. Such underestimation leads to gene content decay curves that depend on the number of genomes included in the subset. Accordingly, the decay curve of subsets of 4 genomes is different from the decay curve of subsets of 2 genomes. **c** The gene content decay curves of the *Bacillus thuringiensis/cereus/anthracis* group are compatible with a scenario of recombination-driven shortening of the terminal branches of the phylogenetic (substitution-based) tree. Based on the tree from Fig. 2a. **d** If the recombination model is used to correct for unobserved variation (fit in Fig. 2c, left panel), overlapping decay curves are obtained. Source data are provided as a Source Data file.

phenomenon affects not only rare genes that are present in a small fraction of the genomes, but also common genes that comprise the core genome (Supplementary Fig. 2). As illustrated by Fig. 1a, b, such a non-overlapping pattern could be easily explained if the lengths of the terminal branches in the phylogenetic trees were systematically underestimated. In more general terms, the curves in Fig. 1c involve two different clocks: the molecular clock, that is used to infer the branch lengths in the phylogenetic tree; and the gene turnover clock, that governs the stochastic process of gene loss and, with it, the decay in the fraction of shared genes. Thus, the absence of overlap among gene sharing curves reflects a non-linear relationship between these two clocks, or more specifically, a delayed onset of the molecular clock relative to the gene turnover clock.

To further explore the differences between the sequence-based molecular clock and the gene turnover clock, we generated an alternative set of 'gene content trees' by inferring branch lengths through phylogenomic analysis while controlling for gene family-specific gain and loss rates (Fig. 2a, b; see Methods section for

details). The tree branch lengths obtained with this approach are congruent with the gene turnover clock, that is, they represent the expected number of gene gains and losses that reach fixation along each branch. For each pair of genomes, we calculated their 'gene content tree distance' by adding up the lengths of the branches that connect both genomes in the gene content tree. Likewise, we used the traditional (substitution-based) phylogenetic trees to calculate 'sequence tree distances'. The non-linear relationship between the molecular clock and the gene turnover clock becomes evident when comparing pairwise distances among leaves in the two sets of trees (Fig. 2c). Sequence divergence only starts building up after a transient period during which genes are gained and lost, although the duration of this transient phase varies across taxa. This observation could be trivially explained if the lengths of the terminal branches in the gene content trees were systematically overestimated, possibly, due to the recent acquisition by single genomes of neutral and deleterious genes that would not reach fixation in the long term. To rule out this possibility, we reconstructed gene content trees from simulated

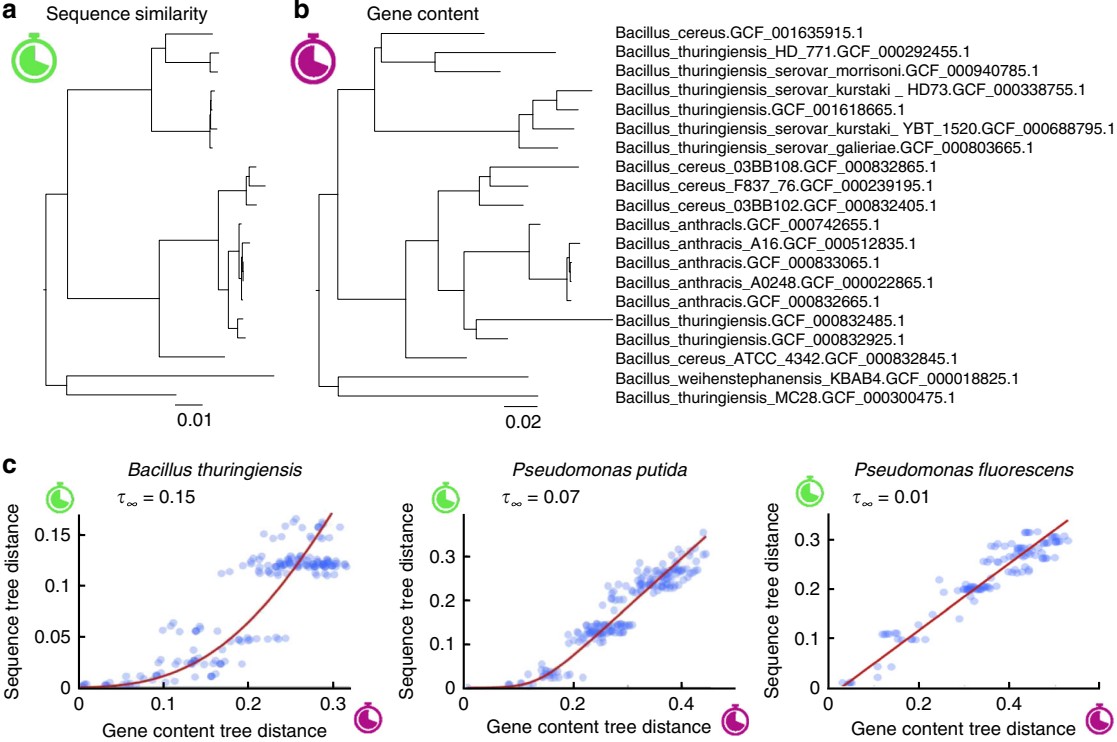

**Fig. 2** Lack of collinearity between sequence and gene content distances supports a recombination-driven delay in the molecular clock. **a** Phylogenetic tree of the *Bacillus thuringiensis/cereus/anthracis* group based on the concatenated alignment of genes shared by all members of the group. **b** The same tree, with branch lengths proportional to the number of gene gain and loss events estimated by phylogenomic analysis. **c** Comparison of pairwise distances between leaves in the sequence similarity and gene content trees, for three representative groups (the left-most plot corresponds to the trees in a and b). The red line is the fit of the recombination barrier model of sequence divergence with a long-term delay in the molecular clock equal to $\tau_\infty$. Source data are provided as a Source Data file.

sets of genomes and verified that the inferred branch lengths were proportional to the times of the simulated evolutionary process. These simulations showed that proportionality between inferred branch lengths and evolutionary time holds for terminal branches, despite the apparent excess of short-lived events involving neutral, deleterious, and fast-turnover genes that characterize such branches (Supplementary Figs. 3 and 4, Supplementary Note 4). To further demonstrate that our results are not significantly confounded by the recent acquisition of deleterious and/or fast evolving genes, we repeated the analysis after removing from the dataset mobile genetic elements and genes with no detected orthologs (ORFans). The results for this pruned dataset were qualitatively and quantitatively similar to those obtained with the complete dataset, demonstrating the lack of an appreciable bias from short-lived gene gains (Supplementary Fig. 5).

**Recombination barrier model explains molecular clock delay.** The observation that early divergence of strains proceeds through a transient stage in which substitutions (effectively) do not accumulate, motivated us to model the dynamics of sequence divergence in the presence of homologous recombination. Rather than focusing on the mechanistic details of recombination, we formulated a phenomenological model for the fraction of loci within a genome that become isolated with respect to another genome from the same original population (see Methods section). Loci that are not affected by barriers to recombination experience periodical gene conversion that reverts them to the population average[22,41]. Our model captures this fact by assuming that recombining loci provide a negligible contribution to the genome-wide sequence divergence.

However, as soon as barriers to recombination are established, the affected loci start diverging from the ancestral population at a rate that is proportional to the substitution rate. As a result, the overall sequence divergence at a given time results from the contribution of all loci that are isolated by barriers to recombination, weighted by the time elapsed since the respective locus crossed the barrier.

We show in the Methods that homologous recombination within a population causes a delay in the molecular clock, such that the average sequence divergence of a genome with respect to a member of the same population grows as

$$\Delta(t) = 2\mu(t - \tau(t)) \qquad (3)$$

where $t$ is the time since the last common ancestor. Mathematically, the delay $\tau(t)$ is a concave and saturating function, the detailed form of which depends on the dynamics of the evolution of recombination barriers (for exact expressions for some simple scenarios, see Supplementary Note 3). For sufficiently long times, this term reaches a constant value $\tau_\infty$, which is the long-term evolutionary delay of the molecular clock induced by homologous recombination. The delay in the molecular clock accounts for the amount of unobserved variation that is erased by gene conversion resulting from homologous recombination during the early phases of divergence from the ancestral population.

A notable consequence of the delay in the molecular clock is that the terminal branches of phylogenetic trees inferred from sequence analysis appear shorter than expected given the actual evolutionary times. Specifically, terminal branches are shortened by a distance $\mu\tau(t_A)$, whereas internal branches are shortened by a distance $\mu\tau(t_A)$-$\mu\tau(t_A)$, where $t_A$ and $t_B$ are consecutive branching times measured from the tips towards the root. Because both $\tau(t_A)$

and $\tau(t_B)$ tend to the same value $\tau_\infty$ as time passes, the recombination-driven delays cancel out at long evolutionary times and deep internal branches remain approximately unchanged.

To test whether the recombination-driven delay is a plausible cause for the lack of linearity between the molecular (sequence divergence) and gene turnover clocks, we used the recombination barrier model to correct the branches of sequence-based trees, accounting for the effects of recombination (see Methods section). Then, we sampled subsets of genomes and reassessed the divergence in gene content as a function of the corrected tree distances. Remarkably, correction of the sequence trees led to gene-sharing decay curves that do not depend on the sample size, as predicted by the theory (Fig. 1d shows a representative case). When extending the same approach to all groups of genomes using taxa-specific delays (see below), we found that the mean separation among the curves obtained for different sample sizes decreased by 30% (permutation test $p = 0.012$; Supplementary Fig. 6). These results are compatible with the existence of a recombination-driven delay in the molecular clock, which causes systematic shortening of the terminal branches of phylogenetic trees built from sequence alignments.

**The dynamics of escape from recombination.** We leveraged the differences between the substitution and gene turnover clocks to obtain quantitative estimates of the recombination-driven delays in different taxa. Starting from (uncorrected) sequence similarity trees (Fig. 2a) and gene content trees (Fig. 2b), we retrieved all pairwise distances among leaves and plotted the distances from the sequence similarity tree against those from the gene content tree (Fig. 2c). The recombination-driven delays were obtained by fitting the recombination barrier model to such plots. To better understand how barriers to recombination spread along the genome, we evaluated several scenarios for the temporal establishment of such barriers (see Methods section). We found that the data in 18 out of the 34 studied groups are best explained by an 'autocatalytic' scenario, in which the barrier spread accelerates as more and more loci become isolated (Supplementary Data 1). The autocatalytic scenario also provides good fits in 13 more groups, although the inclusion of an extra parameter required in this scenario is not statistically justified if the delays are very small. Under the autocatalytic scenario, the fraction of sites susceptible to homologous recombination follows a sigmoidal curve in time, with a relatively sharp transition (with few exceptions) from a state of fully recombining genomes to a state in which all sites freely diverge.

The estimates of the long-term evolutionary delay $\tau_\infty$ range from less than 0.01 to more than 0.5 underreported substitutions per site, with broad variation among prokaryotic groups and sometimes even within the same genus (Fig. 3 and Supplementary Data 1). In approximately half of the groups, sequence divergence appears to be strongly delayed, possibly by the pull of recombination, as indicated by the fact that $\tau_\infty$ is larger than the depth of the sequence similarity tree. In these cases, there are few pairs of genomes that have reached the regime of free (linear) divergence, and the estimation of the upper 95% confidence bound for the long-term evolutionary delay becomes unfeasible. The 5 groups of Firmicutes included in the analysis (covering bacilli, clostridia and streptococci) belong to this category. In contrast, representatives of the genus *Pseudomonas* are characterized by a linear divergence regime, with little or no signs of recombination-driven delay. As already noted, these results cannot be explained by recent gene acquisition or deletion events and are robust to the removal of mobile genetic elements and genes with no orthologs (ORFans) that are likely to evolve under

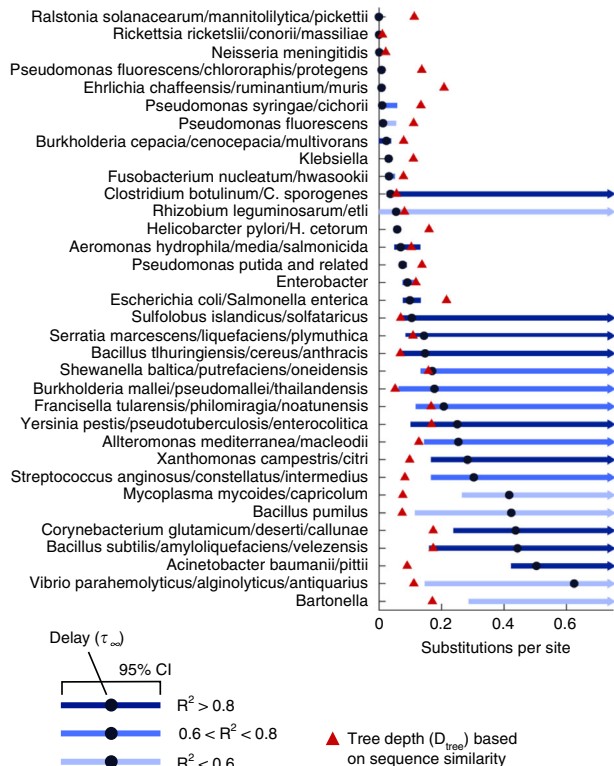

**Fig. 3** Recombination-driven delay ($\tau_\infty$) in the molecular clock in different groups of bacteria. Black circles indicate the best fit of the long-term delay parameter $\tau_\infty$ based on the comparison of sequence similarity and gene content trees. Blue lines show the 95% confidence intervals. Red triangles indicate the total depth of the sequence similarity tree. Values of the delay above or very close to the total tree depth imply that most genomes in a group are strongly bound by homologous recombination; an upper 95% confidence bound cannot be calculated in those cases. Source data are provided as a Source Data file.

a different regime than the rest of the genome (Supplementary Fig. 5).

The magnitude of the recombination-driven delay is tightly linked to the time frame over which sequence divergence takes place. In taxa with little or no delay, variations in the time at which different genes cross the recombination barrier are negligible; from the perspective of divergence times, all genes in these taxa start diverging roughly at the same time (Fig. 4a, left). Conversely, in taxa with lengthy delays, between-gene variations in divergence times can be comparable to the total evolutionary depth of the taxon (Fig. 4a, right). Accordingly, it can be expected that differences in sequence divergence across genes will be larger in taxa with long recombination-driven delays. To test whether genomic data are compatible with this prediction, we first calculated, for a set of 100 nearly universal gene families, the gene family-corrected and taxa-corrected relative evolutionary rates (i.e., the gene-specific substitution rates divided by the gene family-averaged and taxon-averaged substitution rates). It can be shown that the standard deviation of relative evolutionary rates corrected in this way is equal to the coefficient of variation of the times over which genes have been diverging (see Supplementary Note 5). As predicted by the model, there is a significant positive correlation between the recombination-driven delay and the variance of relative evolutionary rates ($R = 0.64$, Pearson's correlation $p < 0.001$; the exact $p$-value associated with the correlation coefficient is calculated in Matlab based on the bivariate distribution). Moreover, when comparing taxa with long

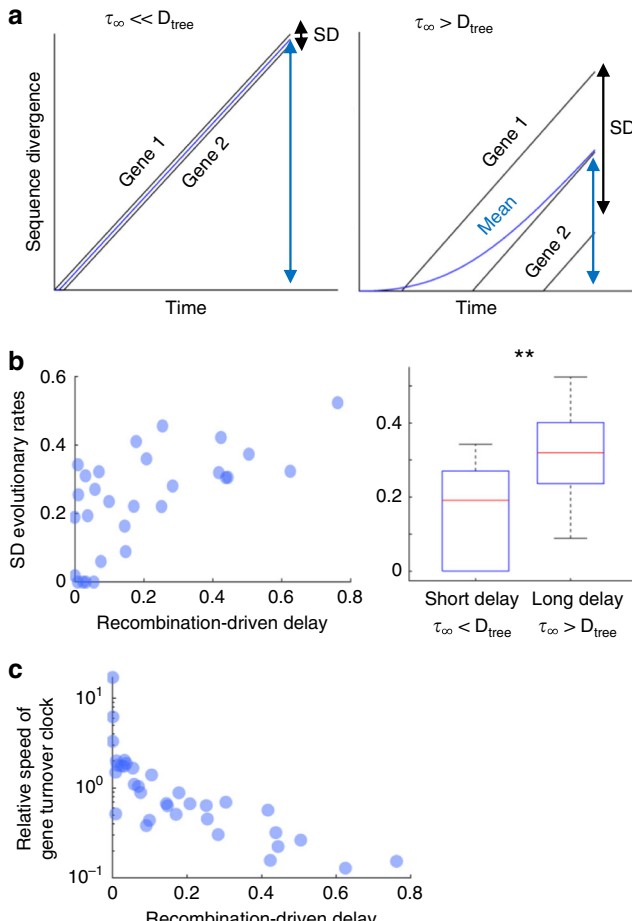

**Fig. 4** : Recombination-driven delay leads to over-dispersion of evolutionary rates within taxa and negatively correlates with gene turnover rate. **a** The divergence of individual genes (black lines) grows linearly after the establishment of barriers to gene conversion, whereas the overall divergence (blue lines) grows non-linearly as the barriers spread across the genome. The mean (blue arrows) and standard deviation (black arrows) of the gene-specific divergences are determined by the values of the recombination-driven delay ($\tau_\infty$) and the tree depth ($D_{tree}$). **b** Standard deviations of the relative evolutionary rates (corrected by gene-wise and taxon-wise rates) as a function of the recombination-driven delay (left). Each data point corresponds to one of the taxa from Fig. 3. Correlation coefficient $R = 0.64$ (Pearson, $p < 0.001$, $n = 34$). On the right, boxplots of the standard deviation of relative evolutionary rates in linearly diverging ($\tau_\infty > D_{tree}$) and strongly delayed taxa ($\tau_\infty \ll D_{tree}$); center line, median; box limits, upper and lower quartiles; whiskers, 1.5× interquartile range; **statistically significant with $p = 0.002$ (two-tailed Student's $T$ test, 32 d.f.). **c** Negative association between the recombination-driven delays and the relative rates of gene turnover with respect to substitutions; Spearman's correlation coefficient rho $= -0.86$ (Spearman's rank-order correlation $p < 0.001$, $n = 34$). Source data are provided as a Source Data file.

and short delays (relative to the evolutionary depth), we found that variance of relative evolutionary rates is significantly greater in the taxa with long delays (Fig. 4b; $p = 0.002$, Student's $T$-test).

To elucidate potential causes for the large variation in the recombination-driven delays found in the data, we searched for associations with genomic and ecological features, such as genome size, number of mobile genetic elements, gene turnover rate, lifestyle (free-living or host-associated), natural competence for transformation, and effective population size (Supplementary

Fig. 7). Among those, we only found a strong negative correlation between the recombination-driven delay and the relative rate of gene turnover with respect to substitutions (Fig. 4c, Spearman's rho $= -0.86$, Spearman's rank-order correlation $p < 0.001$; the exact $p$-value associated with the correlation coefficient is calculated in Matlab based on the bivariate distribution), which indicates that fast gene turnover is associated with a rapid spread of barriers to gene conversion. The statistical power for the analysis of ecological traits (lifestyle and effective population size) was low, and therefore, it cannot be ruled out that these factors contribute to the spread of recombination barriers as well.

In all of the above analyses, we assumed that the equilibrium population diversity sustained by homologous recombination is negligible compared to the divergence in recombination-free regions. If that is not the case, the estimated values of the recombination-driven delays (Fig. 3) become lower bounds for the actual delays (see Supplementary Note 4 for details). Therefore, considering the equilibrium diversity of the population strengthens our conclusion on the existence of a delay in sequence evolution.

## Discussion
Evolution of prokaryotic genomes occurs at two levels: at the sequence level, through substitutions and small indels, and at the genome level, through the transfer and loss of genes and groups of genes[4,42–44]. Whereas evolution at the sequence level is traditionally used to determine phylogenetic relationships among prokaryotes and to assign new genomes to taxonomic groups, it is the gene repertoire (and therefore evolution at the genome level) which determines metabolic capacities, ecological properties and pathogenicity of bacterial strains[45–47]. By studying sequence and gene content divergence in groups of closely related prokaryotes, we found that the onset of sequence evolution is often delayed with respect to genome evolution, although the magnitude of the delay broadly varies across bacterial and archaeal lineages. It has to be stressed that this finding demonstrates distinct dynamics of sequence and gene content divergence and is independent of the difference in the actual rates of these processes. We show that the delay in sequence evolution is likely to result from gene conversion caused by homologous recombination that homogenizes the core genome while, at the same time, many accessory genes can be gained and lost, resulting in gene content divergence and pangenome expansion. These results are fully compatible with our previous findings indicating that archaeal genomes contain a subset of genes that turn over extremely rapidly, before detectable sequence divergence occurs[32]. The delay of sequence divergence caused by homologous recombination provides a plausible mechanistic explanation for the previously observed instantaneous gene turnover in prokaryotes[48]. These results are also compatible with the observation that genes are gained and lost at higher rates on the tips of phylogenetic trees[2].

Notwithstanding the long-term debate on the applicability of the species concept in prokaryotes, several recent studies strongly suggest that speciation does occur in bacteria and is a crucial factor shaping the earth microbiome[17,19,49]. The establishment of barriers to recombination is a pivotal step in the early divergence of closely related prokaryotic strains that eventually leads to speciation[20,28,50]. In the absence of such barriers, sequence divergence is prevented by the cohesive effect of intra-strain homologous recombination; only after the barriers arise and recombination ceases, substitutions start to accumulate at an appreciable rate. By modeling sequence evolution in the presence of gene conversion, we show here that the temporal dynamics for the spread of barriers to recombination directly affects the molecular clock. Specifically, homologous recombination sets

back the molecular clock by an amount of time that, in the long-term, equals the average waiting time for the establishment of recombination barriers. Homologous recombination during early divergence also leads to the compression of the tips of phylogenetic trees. Our model provides a framework for correcting such trees, to make them consistent with the dynamics of gene turnover.

Escape from recombination does not occur in all genes at once, so in principle, a continuum could exist, from closely related strains that can recombine in nearly all shared genes to separate species that are isolated in most if not all genes[25]. The groups of genomes used in this study cover this whole range of divergences, which is what allows us to detect the transition from genomes connected by recombination to freely evolving ones and to quantify the delay in sequence divergence. Furthermore, the autocatalytic (feed-forward loop) dynamics that we detected for the spread of recombination barriers can lead to a relatively sharp transition from strains within a species, in which most genes can recombine, to separate species, in which recombination has ceased in most of the genome. Therefore, the autocatalytic spread of barriers to recombination could contribute to explaining the apparent discrete nature of many prokaryotic species, all the caveats associated with the species concept in the case of prokaryotes notwithstanding[51–53].

The causes that underlie the spread of barriers to recombination are complex and likely involve both genomic and ecological factors. Theoretical models show that barriers can simply emerge if mutations generate diversity faster than recombination erodes it[20–24]. However, it is a matter of debate whether mutation and recombination alone can explain the formation of non-recombining species in nature, and in particular, in prokaryotes[21,22]. The strong, negative correlation between the magnitude of the recombination-driven delay of sequence divergence and the rate of gene turnover that we observed here supports the hypothesis that gene gain and loss accelerate the establishment of barriers to homologous recombination, by promoting niche differentiation[19] and/or by interfering with gene conversion at flanking loci[27–29]. The autocatalytic, sigmoidal dynamics that best describes the fraction of recombination-free loci in our model is consistent with the previous findings indicating that barriers to recombination are initiated by the acquisition of lineage-specific genes and subsequently spread from the vicinity of those genes towards the rest of the genome[25,26]. More generally, our findings imply that gene gain and loss drive speciation in prokaryotes by promoting the establishment of recombination barriers. In this context, speciation appears to be an emergent phenomenon in which a "macroscopic" discontinuity (a sharp barrier to recombination between different species) builds up from a continuous "microscopic" dynamics (the gradual decay in recombination efficiency observed for increasingly divergent strains).

One might argue that the correlation between the rates of recombination barrier establishment and gene turnover does not unambiguously define the causal relationship, and that reverse causality, whereby recombination barriers would accelerate gene turnover, could be a viable alternative to the scenario proposed here. Although we acknowledge this possibility, we argue that the role of gene turnover as a driver of the recombination barrier emergence is far more likely. Indeed, gain and loss of genes creates non-homologous regions between diverging genomes, thus, directly promoting the emergence of homologous recombination barriers. We are unaware of any reciprocal mechanism whereby elimination of homologous recombination would promote gene turnover.

It should be emphasized that our model is purely phenomenological and as such is not contingent on the specific mechanisms behind the emergence of the recombination barriers. In

particular, the model is applicable both to sympatric speciation, whereby prokaryotic strains diverge within the same habitats, and to allopatric speciation when sequence divergence is facilitated by physical (geographical) isolation[54,55].

Recombination-driven delays are indicative of the time frame over which different genes start to diverge during the split of prokaryotic lineages. Lineages with short delays are characterized by low variance in the gene-specific (gene family-corrected and taxon-corrected) evolutionary rates, which implies that most of their genes began diverging over a brief period of time. In contrast, the higher variances observed in lineages with long delays imply that, in those lineages, divergence of genes occurred in an asynchronous way and spanned longer periods of time. To illustrate this point, it has been estimated that the split between *Escherichia* and *Salmonella* took place around 140 million years ago and spanned over 70 million years, which is in a close agreement with our finding that the delay in the *Escherichia*/*Salmonella* group is approximately half of its evolutionary depth. The high variability of the sequence divergence delays across taxa implies that the time required for lineages to split and form new species is strongly taxon-dependent. For taxa with long delays, it appears natural to think of speciation events being driven by the emergence of strong recombination barriers, whereas taxa with short delays would follow a more continuous speciation dynamics, with new species forming as the genomes gradually diverge[21,22]. Although our analysis focused on prokaryotic genomes, the conclusions appear to be general and can be extended to the evolution of other genomes that are substantially affected by horizontal gene transfer and homologous recombination, for example, viruses.

## Methods

**Recombination-driven delay in sequence divergence**. Previous theoretical work has investigated the dynamics of a population subject to mutation and homologous recombination, with recombination rates that decay exponentially with sequence divergence[21,22]. The dynamics of such populations is driven by occasional outbreaks of subclones that accumulate divergence to a point where homologous recombination with the rest of the population ceases. Under the assumptions of Dixit et al.[22], the probability that a subclone reaches a state of nearly irreversible divergence is roughly constant in time and depends on molecular and population parameters, such as the effective population size, mutation rate, and the dependency of the recombination rate on sequence divergence.

Inspired by those results, we adopted a phenomenological approach to model the effect of intra-population homologous recombination on the genetic diversity of a population, taking the probability of escaping recombination (which in our case is not necessarily constant in time) as the main parameter of the model. Under this approach, each genomic region is either subject to periodical recombination with other members of the population or has reached a point where homologous recombination is not possible anymore. In the latter case, the respective genomic region is assumed to have crossed (diverged beyond) the recombination barrier. There was no attempt to explicitly model the molecular mechanisms that underlie barriers to recombination, and therefore, the present model not only captures the main feature of prior divergence-recombination models but also becomes applicable to scenarios in which ecological factors drive the isolation of evolving populations.

Given a pair of genomes, we used a general function $f(t)$ to describe the fraction of each genome that is subject to recombination. The probability that a region of the genome crosses the recombination barrier exactly at time $t$ (where the time starts counting at the last common ancestor of the two genomes) is

$$P(t) = -df/dt \tag{4}$$

Considering that only the regions that have crossed the recombination barrier make a long-term contribution to sequence divergence, and that the number of substitutions in those regions is proportional to the time elapsed since the recombination barrier was crossed, the overall sequence divergence between a pair of genomes becomes

$$\Delta_2(t) = 2\mu \int_0^t (t-u)P(u)du \tag{5}$$

where $\mu$ is the average substitution rate. Integration by parts leads to the final result

$$\Delta_2(t) = 2\mu(t - \tau(t)) \tag{6}$$

where $\tau(t) = \int_0^t f(u)du$ represents a recombination-driven delay in the molecular clock.

To contrast the model with genomic data, we explored three more specific scenarios by assigning explicit functional forms to the rate at which regions of the genome cross the recombination barrier. Thus, we considered a power law time-dependency of the rate $R(t) = \lambda t^y$ (which includes the case of a constant rate); a linear time-dependency plus a constant term $R(t) = \lambda_0 + \lambda_1 t$; and an autocatalytic scenario $R(f) = \lambda_0 - \lambda_1 f$ in which the rate increases as more and more regions of the genome cross the recombination barrier. Given a functional expression for $R(t, f)$, the fraction of the genome subject to recombination is obtained by solving the differential equation $df/dt = R(f, t)f$ (see Supplementary Notes 2 and 3 for further details).

**Model of genome content divergence.** The number of genes, $x$, in a prokaryotic genome was modeled as a stochastic birth-death process, with genome-wide gain rate $P^+$ and loss rate $P^-$[31]. Under this model, the number of genes in a genome is described by the differential equation

$$dx/dt = P^+ - P^- \tag{7}$$

To facilitate comparison of gene content across genomes, each genome was represented by a vector $X$ with elements that assume values of 1 or 0. Each entry represents a gene, i.e., an ATGC-COG, where 1 or 0 indicate the presence or absence, respectively, of that ATGC-COG in the genome. Genome size $x$ is then given by the sum of all elements in $X$.

The number of shared genes in a pair of genomes, or pairwise intersection, $I_2$ is defined as

$$I_2(t) = \langle X_1 \cdot X_2 \rangle \tag{8}$$

where $X_1$ and $X_2$ are vectors that represent the two genomes, the angled brackets indicate averaging over multiple realizations of the stochastic process, and the dot operation stands for a scalar product. The dynamics of pairwise intersections is given by

$$dI_2/dt = 2\langle (dX_1/dt) \cdot X_2 \rangle \tag{9}$$

where we used the fact that both averages are equal $\langle (dX_1/dt) \cdot X_2 \rangle = \langle X_1 \cdot (dX_2/dt) \rangle$. Assuming that genes are acquired from an (effectively) infinite gene pool[56], we have

$$\langle (dX_1/dt) \cdot X_2 \rangle = -P^- \cdot I_2(t)/x \tag{10}$$

and substituting this relation into the differential equation for pairwise intersections of Eq. (9), we obtain

$$dI_2/dt = -2P^- \cdot I_2(t)/x \tag{11}$$

The solution of this equation shows that pairwise genome intersections decay exponentially as

$$I_2(t) = I_2(0)e^{-vt} \tag{12}$$

with decay constant $v = 2P^-/x$. Assuming a molecular clock, the time $t$ can be translated into tree pairwise distance as $D_2 = 2t/t_0$ and the pairwise similarity decays exponentially with the tree distance $D_2$ as

$$I_2(d) = x\,e^{-\lambda D_2} \tag{13}$$

with decay constant $\lambda = t_0 P^-/x$. Note that the ratio $P^-/x$ gives the per-gene loss rate.

The intersection of $k$ genomes, $I_k$, is the number of orthologous genes that are shared by all $k$ genomes. It is formally defined as

$$I_k = \langle \text{intersect}(X_1 \ldots X_k) \rangle \tag{14}$$

Similar to pairwise genome intersections, the time derivative of $k$-intersections is given by

$$dI_k/dt = k\langle \text{intersect}(dX_1/dt \ldots X_k) \rangle = -kI_k(t)P^-/x \tag{15}$$

Solving the differential equation, we obtain an exponential decay for the $k$-intersections

$$I_k(t) = I_k(0)e^{-kP^- t/x} \tag{16}$$

If time is inferred from a tree

$$I_k(D_k) = x\,e^{-\lambda \cdot D_k} \tag{17}$$

where $\lambda$ is proportional to $P_-/x$ and $D_k$ is the sum of branch lengths in the tree that encompasses the $k$ genomes (see Supplementary Note 1 for formal derivation). This expression can be extended to genomes composed of fast and slow evolving genes, and becomes

$$I_k(D_k) = x_1 \cdot e^{-\lambda_1 \cdot D_k} + x_2 \cdot e^{-\lambda_2 \cdot D_k} \tag{18}$$

where $x_1$ and $x_2$ are the average numbers of genes of each class.

**Genomic data.** We used the Alignable Tight Genomic Clusters (ATGC) database[57] to define groups of closely related bacterial and archaeal genomes. By construction, ATGCs are independent of taxonomic affiliation and meet the objective criteria of high synteny and low divergence (synonymous substitution rate dS < 1.5 in protein-coding genes). We selected 36 ATGCs that matched the following criteria: (i) maximum pairwise tree distance is at least 0.1 substitutions per site, to ensure that the ATGC includes well differentiated strains (the presence of closely related strains in an ATGC is guaranteed by construction), such that we could compare the rates of sequence and gene content evolution beyond closely related strains; and (ii) the phylogenetic tree contains more than two clades, such that pairwise tree distances are centered around more than two typical values, to ensure that there were at least two major clusters (generally, species, or at least, clearly differentiated strains), such that the study of that ATGC was informative with regard to the stage of potential incipient speciation. Two of the 36 genome clusters were identified as outliers and were excluded from the dataset. The 34 genome clusters analyzed in this study are listed in Supplementary Data 1. To facilitate computational analysis, we subsampled large ATGCs to keep at most 20 representative genomes per ATGC. ATGC-specific Clusters of Orthologous Genes (ATGC-COGs) were downloaded from the ATGC database and postprocessed to obtain finer grain gene families by reconstructing approximate phylogenetic trees from original ATGC COG alignments and splitting them into subtrees with minimum paralogy. ATGC-specific phyletic profiles were built by registering, as a binary matrix, the presence or absence of each ATGC-COG in each genome within the ATGC (multiple genes from a single genome that belong to the same ATGC-COG were counted once).

**Tree construction.** High-resolution phylogenetic trees based on the concatenated alignments of single-copy core genes were downloaded from the ATGC database[57]. We refer to these trees as sequence similarity-based trees. The phylogenomic reconstruction software Gloome[58] was used to obtain trees based on the gene content similarity among the members of each ATGC. As the input for Gloome, we used the phyletic profiles for the presence or absence of each ATGC-COG, and the sequence similarity-based trees from the ATGC database; options were set to optimize the tree branch lengths under a genome evolution model with 4 categories of gamma-distributed gain and loss rates. This procedure resulted in 2 trees per ATGC, both with the same topology but with different branch lengths (one based on sequence divergence, the other on gene content divergence). All trees were inspected for extremely long and short branches, and clades responsible for such branches (typically 1 or 2 genomes in 5 out of the 34 ATGCs) were manually removed to avoid possible artifacts in the following steps.

**Tree comparison and model fitting.** For each ATGC, we computed all pairwise distances among tree leaves in the sequence similarity-based and gene content-based trees. Then, we compared the observed relationship between both sets of distances with the expectations of the recombination barrier model under four scenarios for the recombination barrier crossing rate (constant, linearly increasing with time, linearly increasing plus a constant term, and proportional to the fraction of the genome that has already crossed, which leads to an autocatalytically accelerated crossing rate). For each scenario, we fitted the model parameters using non-linear least-squares optimization (implemented in Matlab R2018b), with sequence similarity-based tree distances as independent variable and gene content-tree distances as dependent variable. The choice of sequence similarity-based tree distances as the independent variable was motivated by the need to fulfill the assumption of homoscedasticity in the non-linear regression model. In addition, we studied the fit of a heuristic power law model $y = bx^\alpha$. To compare the goodness of fit provided by different models, we calculated the Akaike Information Criterion (AIC) as $\text{AIC} = 2k + n(\ln(2\pi\,\text{RSS}/n) + 1)$, where $k$ is the number of parameters, $n$ is the number of observations, and $RSS$ is the residual sum of squares[59]. The 95% confidence interval for the delay parameter $\tau_\infty$ was obtained by finding the values of $\tau_\infty$ such that the residual sum of squares becomes $\text{RSS} = \text{RSS}^*(1 + 1.96^2/(n-1))$, where $RSS^*$ is the residual sum of squares produced by the best fit of the model[60].

**Correction of tree branch lengths.** To account for unobserved variation, tree branch lengths were corrected by using the autocatalytic barrier spread model with the parameters inferred in the previous step. In that model, the observable divergence increases with time according to the function

$$f(t) = t - \tau_\infty \left(1 - \ln\left(1 + e^{\phi(1 - \xi t/\tau_\infty)}\right) / \ln\left(1 + e^\phi\right)\right) \tag{19}$$

with

$$\xi = \left(1 + e^{-\phi}\right)\ln\left(1 + e^\phi\right)/\phi \tag{20}$$

In the simplest case of an ultrametric tree, the corrected height (the distance from the tip) of a node $i$, $\tilde{h}_i$, can be calculated by applying the inverse function to the original height $h_i$, that is

$$\tilde{h}_i = f^{-1}(h_i) \tag{21}$$

Branch lengths would then be obtained by subtracting the depths of the parent and child nodes. Extending this idea to non-ultrametric trees, we first defined the parental height of a node, $H_i$, as the distance between the node's parent and the tip, that is, $H_i = h_i + b_i$, where $b_i$ is the length of the branch that connects node $i$ to its

parent node. Parental heights were calculated as weighted averages from the tips to the root, such that $H_i = b_i$ for the leaves and

$$H_i = \frac{w_{i1}H_{i1} + w_{i2}H_{i2}}{w_{i1} + w_{i2}} + b_i \qquad (22)$$

for internal nodes. Subindices **i**1 and **i**2 refer to the child nodes of node $i$; weights were computed as $w_i = b_i$ for leaves and $w_i = w_{i1} + w_{i2} + b_i$ for internal nodes (thus, the weight of a node is equal to the total length of the subtree that contains that node as its root plus the length of the branch that connects it to its parent node). Next, corrected values of the parental heights were obtained by computing, numerically (MATLAB R2018b), the inverse function $\tilde{H}_i = f^{-1}(H_i)$. The corrected branch lengths for the tips are simply $\tilde{b}_i = \tilde{H}_i$. Finally, we proceeded from tips to root and obtained the corrected branch lengths associated with internal nodes as

$$\tilde{b}_i = \tilde{H}_i - \frac{\tilde{w}_{i1}\tilde{H}_{i1} + \tilde{w}_{i2}\tilde{H}_{i2}}{\tilde{w}_{i1} + \tilde{w}_{i2}} \qquad (23)$$

where the new weights $\tilde{w}_{i1}$ and $\tilde{w}_{i2}$ were recalculated at each step using the corrected branch lengths.

**Analysis of gene content**. For a set of $k$ genomes that belong to the same ATGC, the gene content overlap was calculated as the number of ATGC-COGs shared by all the genomes divided by the mean number of ATGC-COGs per genome. The total sequence divergence of a set of $k$ genomes was calculated as the sum of all branch lengths in the sequence similarity subtree that results from selecting the corresponding leaves in the whole-ATGC tree. Curves for the temporal decay of the fraction of shared genes were obtained by plotting the gene content overlap against the total sequence divergence for all possible combinations of $k$ genomes within the ATGC. Smooth curves were obtained by fitting a cubic spline model with 5 knots (placed in both extremes and in the 25-percentiles, 50-percentiles, and 75-percentiles of the data $x$-values) using the SLM tool (D'Errico, August 10, 2017 (http://www.mathworks.com/matlabcentral/fileexchange/24443)) in MATLAB R2018b. The mean separation among the curves obtained for different values of $k$ was calculated as

$$S = \left( \frac{\sum_{k=2}^{n-2} \sum_{k'=k+1}^{n-1} \int_{a_{kk'}}^{b_{kk'}} (f_k(x) - f_{k'}(x))^2 dx}{\sum_{k=2}^{n-2} \sum_{k'=k+1}^{n-1} (b_{kk'} - a_{kk'})} \right)^{1/2} \qquad (24)$$

where $n$ is the number of genomes in the ATGC, $f_k$ and $f_{k'}$ are the curves that result from fitting the spline model to sets of $k$ and $k'$ genomes, respectively, and $a_{kk'}$ and $b_{kk'}$ are the bounds of the $x$-axis interval in which both $f_k$ and $f_{k'}$ are defined[61]. A value of $S = 0$ corresponds to the exact coincidence of the curves for all values of $k$, which is expected in the absence of evolutionary delays. To assess whether correction of branch lengths for unobserved variation reduces the separation among curves, we calculated the relative change in separation as $(S_{\mathrm{original}} - S_{\mathrm{corrected}})/\max(S_{\mathrm{original}}, S_{\mathrm{corrected}})$, which takes the value of 1 when correction leads to complete collapse, positive values $\leq 1$ when correction reduces separation, and negative values $\geq -1$ when correction increases separation among curves. The statistical significance of the relative change in separation was assessed using a permutation test that involved calculating the median relative change across ATGCs and comparing it with $10^6$ randomized datasets in which the 'original' and 'corrected' labels were randomly reassigned in each ATGC.

**Quantification of evolutionary rates**. For the analysis of evolutionary rates, we focused on a previously published list of 100 nearly universal gene families[62], defined as clusters of orthologous genes or COGs[63]. To minimize possible confounding effects due to paralogy, we identified all the ATGC-COGs that match any of the universal COGs and restricted the analysis to those COGs that are represented by a single ATGC-COG in at least 30 of the 34 analyzed ATGCs. Multiple sequence alignments for the selected ATGC-COGs were downloaded from the ATGC database and processed to extract all pairwise distances (Nei-Tamura method). Only index orthologs from the ATGC database, i.e., a single sequence per ATGC-COG per genome, were included. For each ATGC-COG, pairwise distances between sequences were plotted against pairwise distances in the phylogenetic tree, and a linear regression model with zero intercept was applied to obtain the relative evolutionary rate of the ATGC-COG with respect to the ATGC average. To minimize the impact of rare instances of gene replacement, which manifest as a non-linear relationship between sequence and tree pairwise distances, we discarded the ATGC-COGs with the fit to the regression model $R^2 < 0.9$. To account for COG-specific evolutionary rates, the relative evolutionary rate of each ATGC-COG was divided by the mean of all ATGC-COGs that match the same COG. The result is the ATGC-COG relative evolutionary rate, that is, the ATGC-COG evolutionary rate corrected by COG-wise and ATGC-wise averages. For each ATGC, the dispersion of evolutionary rates was quantified as the standard deviation of the relative evolutionary rates of its ATGC-COGs.

**Reporting summary**. Further information on research design is available in the Nature Research Reporting Summary linked to this article.

## Data availability

The datasets generated and/or analyzed in the course of this study are available from the corresponding authors upon request. The source data underlying Figs. 1cd, 2abc, 3, 4bc, and Supplementary Figs. 1, 2, and 5–7 are provided as a Source Data file.

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

## Acknowledgements

The authors thank Koonin group members for helpful discussions. The authors' research is supported by intramural research program funds of the National Institutes of Health (National Library of Medicine). J.I. is funded by the "Ramón y Cajal Program" from the Spanish Ministry of Science and Innovation and by the "Severo Ochoa Program for Centres of Excellence in R&D" from the Agencia Estatal de Investigación of Spain (grant SEV-2016-0672 (2017–2021) to the CBGP). This work utilized the computational resources of the NIH HPC Biowulf cluster. (http://hpc.nih.gov).

## Author contributions

J.I. and I.S. performed research; J.I., Y.I.W., E.V.K. and I.S. analyzed the data; J.I., E.V.K. and I.S. wrote the paper that was edited and approved by all authors.

## Competing interests

The authors declare no competing interests.
