## [Peer Review File · Nature Communications]

Reviewer #1 (Remarks to the Author):

The manuscript addresses a topic that is fundamental, if of somewhat specialized interest, in molecular evolution -- the discrepant patterns of evolutionary rates at the genic level (substitutions) versus the genomic level (gene gains and losses).

It has previously been observed that these two different types of molecular clocks are not the same, nor even linearly related; and several mechanisms have been proposed before, and are reiterated here. Nonetheless this work provides both a synthetic empirical overview of these patterns within archaea and bacteria, and, more importantly, a model based on recombination-driven delays in substitution rates, for why these discrepancies might arise. The main idea of the model is that, due to gene conversion, sequence divergence is initially slower than gene loss/gains, until a transient period is passed and sequence divergence subsequently accelerates. This model makes perfect logical sense as an explanation for the basic empirical pattern being discussed.

It seems this manuscript has already seen several rounds of review and cross-review, and so I will limit my comments to assessing the potential concerns that have been raised, and whether they have been mitigated, in my view. And finally I will comment briefly on the value of the work as a whole, in my view.

The most (in fact, only remaining) pressing concern was a possible artifactual reason that might explain the observed differences in rate of recent evolution in genes (substitutions) versus genomes (gene loss/gains) -- that the lengths of terminal branches might be overestimated due to recent acquisition of neutral and deleterious genes that will not eventually fix. The authors have now addressed this concern with rigor, in my view -- noting that gene content differences are measured in expected counts of losses/gains, not observed counts. The authors have performed several control experiments (re-analyses of subsets of the data, as well as simulations) to show that this way of measuring gene content variation is not subject to the artifact that was the source of the referee's original (and potentially valid) concern.

The other concerns that remain from referee #1 and referee #3 seem to be largely ones of interpretation and presentation, and these too seem to have been considered and addressed in this revised version.

My only remaining concern has to do with the interpretation of the model as phenomenological versus mechanistic explanation for the empirical patterns. At several points, in response to referees, the authors point out that the model is purely phenomenological -- which is important, because it therefore equally well describes recombination barriers arising from any processes, whether ecological or from molecular processes of gene conversion. And yet, at the same time, the authors describe their model as "mechanistic" in the abstract of the manuscript. (So what it is, after all, phenomenological or mechanistic?) Moreover, the model is, in places, given a mechanistic interpretation, in the sense that variation in

discrepancies between clocks across lineages are interpreted in terms of the model -- eg, 18 of the 34 microbial groups are explained by an "autocatalytic" scenario, which is closely tied to a mechanistic description of how sites susceptible to homologous recombination changeover time. As a reader I am left unclear how the model is meant to be interpreted, eg does

it make predictions that could be falsified by empirical measurements of molecular recombination processes (eg the rate of recombination as a function of identity of flanking sequences)?

I am supportive of publication largely as the it stands. But I recommend the authors consider once more what parts of their modelling is phenomenological, and what parts have obvious interpretations as driven by mechanisms of gene conversion processes -- and that they revise their text to be more consistent on this point.

--Joshua Plotkin

MINOR POINT: line 235: "the standard deviation of evolutionary rates ... is equal to the coefficient of variation of the times over which genes have been diverging". I am mystified by this claim, because the two quantities seem to have different units. The first has the unit of evolutionary rates (1/times), and the second quantity is unitless -- so how can they be equal, in general?

Authors' response:

We appreciate Dr. Plotkin's positive and constructive review.

In the revision, we emphasize that the model is essentially phenomenological but has mechanistic implications.

With regard to the minor comment of Dr. Plotkin, we modified the text to explain that, as calculated in our work, both quantities are dimensionless.